# RMP-SAM: Towards Real-Time Multi-Purpose Segment Anything

**Shilin Xu**[1,4†*]**, Haobo Yuan**[2,3†]**, Qingyu Shi**[1]**, Lu Qi**[3]**, Jingbo Wang**[4]**, Yibo Yang**[5]**, Yining Li**[4]**,
**Kai Chen**[4]**, Yunhai Tong**[1]**, Bernard Ghanem**[5]**, Xiangtai Li**[2,4‡]**, Ming-Hsuan Yang**[3,6]
[1]Peking University, [2]Nanyang Technology University, [3]UC, Merced
[4]Shanghai AI Laboratory, [5]KAUST, [6]Google Research
xushilin@stu.pku.edu.cn, xiangtai94@gmail.com

## ABSTRACT

Recent segmentation methods, which adopt large-scale data training and transformer architecture, aim to create one foundation model that can perform multiple tasks. However, most of these methods rely on heavy encoder and decoder frameworks, hindering their performance in real-time scenarios. To explore real-time segmentation, recent advancements primarily focus on semantic segmentation within specific environments, such as autonomous driving. However, they often overlook the generalization ability of these models across diverse scenarios. Therefore, to fill this gap, this work explores a novel real-time segmentation setting called real-time multi-purpose segmentation. It contains three fundamental sub-tasks: interactive segmentation, panoptic segmentation, and video instance segmentation. Unlike previous methods, which use a specific design for each task, we aim to use only a single end-to-end model to accomplish all these tasks in real-time. To meet real-time requirements and balance multi-task learning, we present a novel dynamic convolution-based method, Real-Time Multi-Purpose SAM (RMP-SAM). It contains an efficient encoder and an efficient decoupled adapter to perform prompt-driven decoding. Moreover, we further explore different training strategies and one new adapter design to boost co-training performance further. We benchmark several strong baselines by extending existing works to support our multi-purpose segmentation. Extensive experiments demonstrate that RMP-SAM is effective and generalizes well on proposed benchmarks and other specific semantic tasks. Our implementation of RMP-SAM achieves the optimal balance between accuracy and speed for these tasks. The code is released at https://github.com/xushilin1/RAP-SAM

## 1 INTRODUCTION

Recent advancements in computer vision, driven by transformer architectures (Dosovitskiy et al., 2021; Carion et al., 2020; Wang et al., 2023b), focus on creating a single large-scale model versatile enough to handle different tasks like detection and segmentation with a general-purpose design. Several studies utilizing the same architectures (Wang et al., 2023a; Yan et al., 2023; Jain et al., 2023) have shown superior performance across various tasks. Enjoying the benefit of the large-scale model capacity and training data, the general-purpose models demonstrate strong performance on versatility for specific applications (Kirillov et al., 2023; Jia et al., 2021). For image and video segmentation at the semantic level, numerous studies (Cheng et al., 2022; Zhang et al., 2021; Li et al., 2022d; 2023e; Jain et al., 2023) employing a universal design have surpassed previous models tailored for specific image and video segmentation tasks. The Segment Anything Model (SAM) (Kirillov et al., 2023) is a leading approach in interactive segmentation, offering a unified model for various interactions such as point, bounding box, mask, and text input.

---

[*]Work done during an internship at Shanghai AI Lab
[†]Equal contribution
[‡]Corresponding author

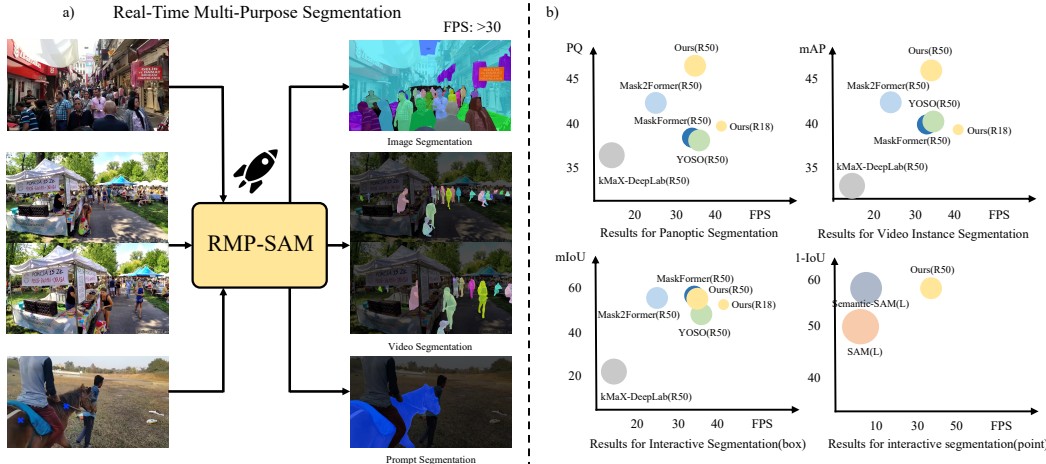

Figure 1: We present real-time multi-purpose segmentation to segment and recognize objects for image, video, and visual prompt inputs. In addition to benchmarking, we propose a simple yet effective baseline, named RMP-SAM, which achieves the best performance and speed trade-off among three different tasks. The larger dot indicates more parameters.

However, most of these studies face challenges in achieving real-time performance on devices with limited computing resources due to their *heavy encoders* or *cascaded decoders*. Despite their high-performance capabilities and versatility, this limitation hinders the practical use of these advanced architectures in vision applications, such as editing tools on edge devices. In contrast, several studies (Zhao et al., 2018; Hu et al., 2023) focus on real-time segmentation using specific designs (e.g., efficient decoder fusion) but are often tailored to a specific application trained on a particular dataset (e.g., COCO) or restricted to similar tasks like video semantic segmentation. To our knowledge, no study investigates real-time, multi-purpose segmentation, where a single model can perform universal segmentation tasks, including image, video, and SAM-like interactive segmentation. For example, given a video input, we want to create a tool to perform various segmentation tasks on the device, including interactive segmentation with visual cues, labeling the masks, and tracking the masks.

Inspired by the abovementioned challenges, we focus on real-time multi-purpose segmentation and efficiently investigate the novel issue of training versatile models. We pose a critical inquiry: Given the limited computational resources and model capacity, how can we develop an efficient, multi-purpose segmentation model? This entails creating a single model capable of segmenting, tracking, and classifying each pixel in real-time, similar to performing interactive segmentation similar to SAM. We present an illustration figure in Fig. 1(a).

First, we explore this problem by benchmarking existing methods. Specifically, we use the COCO (Lin et al., 2014) and YouTube-VIS (Yang et al., 2019) for joint co-training, employing identical hyperparameters. Beyond the object queries used for image and video segmentation in prior research, our approach incorporates visual prompt queries for interactive segmentation. However, none of these works can achieve the best accuracy and speed trade-off on three tasks using one model. This is because: 1) No previous works explore joint co-training for segmentation for different purposes. Thus, extensive experiments are needed to gain a better understanding of meta-architecture. 2) Interactive and semantic-level segmentation needs better strategies to balance each other. 3) Under the real-time setting, several previous architectures (Yan et al., 2023; Li et al., 2022c) for multi-task learning may not be practical.

To solve these issues, we present a real-time multi-purpose segment anything model (RMP-SAM). It contains an efficient encoder and shared decoder with multiple queries as inputs. Unlike previous works, we avoid cascaded transformers encoder layers and pixel-wise cross-attention decoder (Cheng et al., 2022). Instead, we propose a pooling-based dynamic convolution to replace per-pixel cross-attention to achieve better accuracy and speed trade-off. We represent all three tasks using queries with shared dynamic convolution operation. According to different outputs, we can divide the segmentation tasks into two types: one for class-agnostic masks (SAM-like) and the other for class-

Table 1: Real-Time Segmentation Settings. Our multi-purpose segmentation supports more tasks in one framework.

| Property | Image Seg | Video Seg | SAM-Like | SAM-2-Like | Ours: Real-Time Multi-Purpose Seg |
|---|---|---|---|---|---|
| Image Masks | ✓ | ✗ | ✓ | ✓ | ✓ |
| Video Masks | ✗ | ✓ | ✗ | ✓ | ✓ |
| Interactive | ✗ | ✗ | ✓ | ✓ | ✓ |
| Semantic Labels | ✓ | ✓ | ✗ | ✗ | ✓ |
| Multitasks | ✗ | ✗ | ✗ | ✗ | ✓ |

Table 2: Comparison of Segmentation Methods. Our proposed RMP-SAM supports various segmentation tasks and performs them in real-time.

| Methods | SS | PS | VIS | Interactive | Multi-Task in One Model | Real Time |
|---|---|---|---|---|---|---|
| ICNet (Zhao et al., 2018) | ✓ | ✗ | ✗ | ✗ | ✗ | ✓ |
| Bi-Seg (Yu et al., 2018) | ✓ | ✗ | ✗ | ✗ | ✗ | ✓ |
| YOSO (Hu et al., 2023) | ✓ | ✓ | ✗ | ✗ | ✗ | ✓ |
| Mobilie-VIS (Zhang et al., 2024) | ✗ | ✗ | ✓ | ✗ | ✗ | ✓ |
| SAM (Kirillov et al., 2023) | ✗ | ✗ | ✗ | ✓ | ✗ | ✗ |
| Mask2Former (Cheng et al., 2022) | ✓ | ✓ | ✗ | ✗ | ✗ | ✗ |
| Video K-Net (Li et al., 2022d) | ✗ | ✓ | ✓ | ✗ | ✗ | ✗ |
| OneFormer (Jain et al., 2023) | ✓ | ✓ | ✗ | ✗ | ✓ | ✗ |
| RMP-SAM (Ours) | ✓ | ✓ | ✓ | ✓ | ✓ | ✓ |

specific masks (contains one mask and one class label). Two types have different goals and will conflict during the multi-task learning process. Thus, we further explore four different architectures in the case of various decoder designs. In particular, we find a single decoder with multiple queries for different tasks can achieve good enough results and best speed and accuracy trade-off. Moreover, we present an effective dual adapter design that better adapts shared knowledge of the same decoder. The dual adapter contains an asymmetric design to better transfer shared features. To the best of our knowledge, RMP-SAM is the *first* real-time multi-purpose segmentation model that can achieve the best speed and accuracy trade-off on four different tasks, as shown in Fig. 1(b). Moreover, due to the semantic level and interactive level segmentation ability, our model can create more new applications, such as interactive video segmentation. In addition, we also verify our model in the SAM-like setting by co-training with SAM-data, it also achieves comparable results. Extensive experiments show the effectiveness of our design. The main contributions of this work are:

- We introduce real-time multi-purpose segmentation, a novel multi-task segmentation aiming to segment objects for real-time image, video, and interactive inputs.

- We benchmark several real-time transformer-based segmentation approaches for the new settings.

- We present a simple yet fast baseline named RMP-SAM. It contains a lightweight feature extractor, a unified decoder, and two asymmetric adapters.

- Extensive experiments demonstrate that RMP-SAM achieves the best speed and accuracy trade-off in the proposed benchmark and regular real-time semantic and panoptic segmentation benchmarks. We also show scalability across datasets and application demos.

## 2 REAL-TIME MULTI-PURPOSE SEGMENTATION

**Problem Settings.** Our proposed real-time multi-purpose segmentation contains three visual inputs: image, video, and visual prompts (boxes and points). It predicts the corresponding masks, labels, and instance IDs for the content within the given input. Basically, we adopt the definition of panoptic segmentation Kirillov et al. (2019) for image inputs since it covers all concepts in the scene. For video inputs, we segment and track each foreground instance (video instance segmentation). Given visual prompts, we follow the SAM-like Zhang et al. (2023a); Zhou et al. (2023); Shu et al. (2023); Xiong et al. (2023) setting for interactive segmentation, performing class-agnostic segmentation according to the user's inputs. Although recent advancements, such as SAM-2 (Ravi et al., 2024), have attempted to unify the interactive segmentation task for both image and video inputs, our framework emphasizes the importance of object semantics. It incorporates multi-task learning in a single shot. Combined with interactive segmentation and video segmentation, we can also achieve interactive

video segmentation as video object segmentation. As shown in Tab. 1 and Tab. 2, we compare previous segmentation methods with and indicate that our method unifies different segmentation tasks within a single framework. This approach enables more flexible usage across various segmentation tasks, particularly regarding input modalities and predictions.

**Datasets.** For benchmarking, we use the well-known COCO dataset (Lin et al., 2014) for panoptic and interactive segmentation. During the training of interactive segmentation, the boxes derived from the ground truth, along with points randomly sampled from the masks, serve as visual prompts. For testing, we obtain the center point of the mask as a visual prompt. For video segmentation, we adopt the widely used YouTube-VIS 2019 dataset (Yang et al., 2019) for training. We *do not* use the SAM data for our benchmarking. There are several reasons: (1) SAM data is not as well annotated as the COCO dataset. (2) We only focus on object-level and scene-level segmentation, while SAM data contain more granularity. (3) SAM data has no semantics, while our goal is to predict semantic labels for real applications. To fairly compare with SAM-like models, we adopt the SAM-data training for specific setting, as shown in Sec. 4.1.

## 3 PROPOSED METHOD

First, we revise panoptic segmentation, video instance segmentation, and interactive segmentation under the unified formulation. When combined with a query-based segmentation model, object queries can represent all entities within the scene, encompassing image-level objects, video-level tracked objects, and prompt-based specific objects. For panoptic segmentation (Kirillov et al., 2019), given an input image $I \in \mathbb{R}^{H \times W \times 3}$, the objective is to generate a group of masks $\{y_i\}_{i=1}^{N} = \{(m_i, c_i)\}_{i=1}^{N}$ where $c_i$ denotes the class label of the binary mask $m_i$ and $N$ is the number of queries, $H \times W$ indicates the spatial dimensions. For video instance segmentation (Yang et al., 2019), given a video clip input $V \in \mathbb{R}^{T \times H \times W \times 3}$, where $T$ denotes the number of frames, the process aims to generate an instance mask $\{y_i\}_{i=1}^{N} = \{(m_i, c_i, d_i)\}_{i=1}^{N}$, where $N$ represents the number of the tube masks $m_i \in \{0, 1\}^{T \times H \times W}$, $c_i$ indicates the class label of the tube $m_i$, and $d_i$ identifies the instance ID for each instance. For SAM-like segmentation (Kirillov et al., 2023), both the image $I$ and visual prompts $P \in \mathbb{R}^{K \times 2}$ (point prompts are used here) are taken as inputs. These prompts, which include points and boxes, lead to the outputs of corresponding binary image masks $\{y_i\}_{i=1}^{K} = \{m_i \in \{0, 1\}^{H \times W}\}_{i=1}^{K}$, where $K$ denotes the number of visual prompts. Each visual prompt is encoded into one object query, which naturally can be the input of the decoder.

### 3.1 RMP-SAM ARCHITECTURE

**Overall Architecture.** As shown in Fig. 2, our RMP-SAM is built upon a simple encoder and decoder architecture that can input images, videos, and visual prompts. Following SAM, we utilize the prompt encoder to convert visual prompts into queries. These visual prompts and object queries share the same decoder for the efficiency of computation and parameter size. However, the goals of the two types of queries are different: the former pays more attention to the local details of the visual prompts, while the latter considers the scene and temporal features. To better balance the results for interactive segmentation and image/video segmentation, we design a prompt adapter and an object adapter at the end of the decoder.

**Lite Feature Extractor.** Due to computation cost limitations, we avoid using previous methods (Cheng et al., 2022; Li et al., 2024b) that involve large backbones and heavier encoders. As required by real-time constraint, we investigate lightweight backbones such as ResNet18 (He et al., 2016) and TopFormer (Zhang et al., 2022). We incorporate a feature pyramid network with deformable convolution to enhance representation alignment and fuse multi-scale features in real-time. When dealing with video input, we extract the spatial-temporal features. For simplicity, we use $F_{img} \in \mathbb{R}^{\frac{H}{4} \times \frac{W}{4} \times d}$ for image inputs and $F_{vid} \in \mathbb{R}^{T \times \frac{H}{4} \times \frac{W}{4} \times d}$ for video inputs.

**Unified Dynamic Convolution Decoder.** The goal of the decoder is to refine the object query. However, many of these methods (Cheng et al., 2021b; 2022), are not feasible for real-time settings due to their reliance on heavily cascaded layers and pixel-wise cross-attention mechanisms. In contrast, our method uses a pooling-based dynamic convolution framework (Zhang et al., 2021; Li et al., 2022d) to enhance the efficiency of the decoder. Given object query $Q \in \mathbb{R}^{N \times d}$ and prompt query $P \in \mathbb{R}^{K \times d}$, the masks $M_i$ are obtained by dot product with $F_{img}$ or $F_{vid}$. Here, $i$ represents the decoder stage

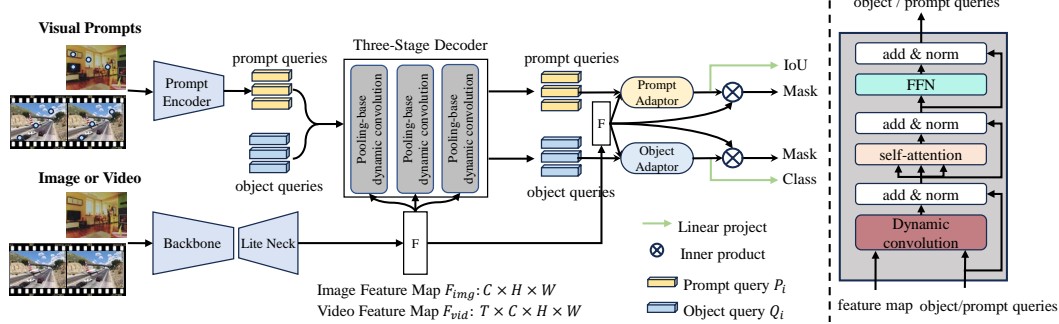

Figure 2: **RMP-SAM overview.** Our method contains three visual inputs: image, video, and visual prompts. Utilizing positional encoding, we generate prompt queries from these visual prompts. The learnable object queries, prompt queries, and the feature map $F$ are directed to the multi-stage decoder. This process generates multi-stage predictions and refined queries. These refined queries engage in cross-attention with $F$, resulting in the final prediction.

index, $N$ is the total number of object queries, and $K$ is the number of visual prompts. According to the inputs, the masks can be image masks for panoptic segmentation $M_i^{pan} \in \mathbb{R}^{N \times \frac{H}{4} \times \frac{W}{4}}$ and interactive segmentation $M_i^{iter} \in \mathbb{R}^{K \times \frac{H}{4} \times \frac{W}{4}}$ or tube masks $M_i^{tube} \in \mathbb{R}^{N \times T \times \frac{H}{4} \times \frac{W}{4}}$. Then, we can obtain the corresponding query features $X_i^{pan} \in \mathbb{R}^{N \times d}$, $X_i^{iter} \in \mathbb{R}^{K \times d}$, $X_i^{tube} \in \mathbb{R}^{N \times d}$ via mask pooling:

$$X_i^{pan} = \sum_u^W \sum_v^H M_{i-1}^{pan}(u, v) \cdot F_{img}(u, v). \tag{1}$$

For video tasks, we group the spatial-temporal features as follows:

$$X_i^{tube} = \sum_t^T \sum_u^W \sum_v^H M_{i-1}^{tube}(t, u, v) \cdot F_{vid}(t, u, v), \tag{2}$$

where $u, v$ are the indices of spatial location, $t$ is index of input frame. Since the dynamic convolution performs similarly for each task, we use panoptic segmentation $X_i^{pan}$ as an illustration. For interactive segmentation, we adopt the same equation (Equ. 1). In our implementation, the learned query is shared for both $Q_i^{pan}$ and $Q_i^{tube}$.

In particular, we adopt the design of previous methods (Zhang et al., 2021; Sun et al., 2021; Li et al., 2022c) to refine input queries $Q_i^{pan}$ with features $X_i^{pan}$ which are grouped from their masks,

$$\hat{Q}_i^{pan} = \text{DynamicConv}(X_i^{pan}, Q_{i-1}^{pan}), \tag{3}$$

where the dynamic convolution uses the query features $X_i^{pan}$ to generate parameters to weight input queries $Q_{i-1}^{pan}$. Specifically, DynamicConv uses input query features $X_i^{pan}$ to generate gating parameters via a multilayer perceptron (MLP) and multiply back to the original query input $Q_{i-1}^{pan}$. We adopt the same design (Zhang et al., 2021; Sun et al., 2021) by learning gating functions to update the refined queries. The DynamicConv operation is based on:

$$\hat{Q}_i^{pan} = \text{Gate}_x(X_i^{pan})X_i^{pan} + \text{Gate}_q(X_i^{pan})Q_{i-1}^{pan}, \tag{4}$$

where $\text{Gate}$ is implemented with a fully connected layer followed by Layer Norm and a sigmoid layer. We adopt two different gate functions, including $\text{Gate}_x$ and $\text{Gate}_q$. The former is to weigh the query features, while the latter is to weigh corresponding queries. Next, we adopt one self-attention layer with feed-forward layers (Vaswani et al., 2017; Wang et al., 2021) to learn the correspondence among each query and update them accordingly. This operation leads to the full correlation among queries, shown as follows:

$$Q_i = \text{FFN}(\text{MHSA}(\hat{Q}_i) + \hat{Q}_i), \tag{5}$$

where MHSA means Multi Head Self Attention, FFN is the Feed Forward Network commonly used in the current version of Transformers (Carion et al., 2020; Dosovitskiy et al., 2021). We adopt several feed-forward layers on $Q_i^{pan}$ for mask classification and directly output the class scores. Then, we

perform the inner product between the learned queries $Q_i^{pan}$, and image features $F_{vid}$ and $F_{img}$. The only difference for the video segmentation task is that we predict the tube mask rather than only a single image mask. The entire process (Equ. 1, Equ. 2, Equ. 3 and Equ. 5) is repeated three times in total.

Moreover, we also explore various decoupled designs, as shown in Fig. 3. In particular, as previous works suggest, we also explore decoupled decoders and their corresponding adapters. Via our detailed experiments, we do not find extra gains in the cases of decoupled decoders that introduce more computation and parameter costs. Please refer to Sec. 4.2 for the detailed decoder design.

**Light-Weight Decoupled Adapter.** After the shared decoder, we also add two lightweight adapters, $A_{obj}$ and $A_{prompt}$ to adapt the shared decoder's knowledge better. In particular, we use the asymmetric design. $A_{obj}$ uses the same dynamic convolution design to refine the object query further, while $A_{prompt}$ uses pixel-wise cross-attention design. Our key insights are: (1) The interaction between the whole scene feature and object queries is necessary for panoptic and video segmentation but not for interactive segmentation. The visual prompts provide strong positional information, and they pay more attention to local details features. (2) Despite the decoder being shared, the details are still missing for interactive segmentation due to the pooling effect in Equ. 1, where we even find inferior results after adopting a pooling-based adapter. More detailed adapter design can be found in Sec. 4.2.

## 3.2 TRAINING, INFERENCE, AND EXTENSION

**Joint Image and Video Segmentation Co-training.** Our goal is to train all segmentation tasks only once jointly. All training targets are one entity label and mask for all three cases. The entity can be thing, stuff, class-agnostic masks, and corresponding labels. Note that the instance masks with the same ID $d$ form the tube masks. During training, we apply Hungarian matching between the predicted and ground-truth entity masks to assign object queries to video/image entities and then supervise their predicted masks and classification. To unify label taxonomies across datasets, we replace the learnable classifier with CLIP text embeddings following ViLD (Gu et al., 2021).

Following previous works (Cheng et al., 2022; Li et al., 2023e), the final loss function is given as $L = \lambda_{cls} L_{m\_cls} + \lambda_{ce} L_{m\_ce} + \lambda_{dice} L_{m\_dice}$. Here, $L_{m\_cls}$ is the Cross-Entropy (CE) loss for mask classification, and $L_{m\_ce}$ and $L_{m\_dice}$ are mask Cross-Entropy (CE) loss and Dice loss (Milletari et al., 2016; Wang et al., 2020) for segmentation, respectively. By default, we set $\lambda_{cls} = 2, \lambda_{ce} = 5, \lambda_{dice} = 5$.

**Inference.** We follow the same inference procedure of Mask2Former (Cheng et al., 2022) for image segmentation. In panoptic segmentation, we merge the prediction of things and stuff according to the sorted scores. For video instance segmentation, we use query matching rather than introducing extra tracking components to generate instance ID, following previous work (Li et al., 2022d; Huang et al., 2022; Li et al., 2023e). We adopt a near-online method for inference. We follow the original SAM (Kirillov et al., 2023) for interactive segmentation tasks by providing box and point prompts and obtaining the binary masks. All the parameters are *shared* across three different tasks.

**Extension and Application.** Since our model can perform various segmentation tasks, Our method can be extended to more datasets and tasks, including semantic segmentation on ADE20k (Zhou et al., 2017) or video panoptic segmentation on VIPSeg (Miao et al., 2022). Moreover, our method can support prompt-driven video object segmentation by combining interactive segmentation with a video instance segmentation framework. We put the details of the extension and application for RMP-SAM in the appendix (Sec. 7.1).

## 4 EXPERIMENTS

**Datasets.** Our benchmark uses COCO (Lin et al., 2014) and YouTube-VIS 2019 (Yang et al., 2019) datasets for benchmarking. COCO-SAM is constructed using ground truth boxes and points randomly sampled from ground truth masks as visual prompt inputs. In addition, to verify the effectiveness and generality of RMP-SAM, we also use other datasets, including ADE-20K (Zhou et al., 2017) and VIP-Seg dataset (Miao et al., 2022) in Sec. 4.1.

**Evaluation Metrics and Devices.** For image segmentation, we adopt panoptic quality (PQ), which can be further decomposed into the segmentation quality (SQ) and the recognition quality (RQ). For

Table 3: Real-Time Multi-Purpose Segmentation benchmark. Our proposed RMP-SAM achieves the best accuracy and speed trade-off on three tasks.

| Method | Backbone | COCO-Panoptic | | | | COCO-SAM mIoU | YouTube-VIS 2019 mAP | FLOPs | Parameters | FPS |
| | | PQ | SQ | PQ_{th} | PQ_{st} | | | | | |
|---|---|---|---|---|---|---|---|---|---|---|
| Mask2Former | R18 | 35.6 | 77.5 | 39.0 | 30.3 | 54.7 | 38.6 | 89.8G | 18.6M | 31.2 |
| MaskFormer | R18 | 31.0 | 76.0 | 33.2 | 27.7 | **55.6** | 34.1 | 79.7G | 18.5M | 38.0 |
| kMaX-DeepLab | R18 | 27.8 | 71.0 | 30.3 | 24.1 | 16.9 | 23.1 | 87.1G | 18.7M | 15.0 |
| YOSO | R18 | 31.6 | 76.8 | 36.4 | 24.4 | 45.7 | 31.4 | 57.3G | 18.7M | 41.0 |
| **RMP-SAM(Ours)** | R18 | **39.9** | **78.6** | **43.3** | **34.8** | 52.7 | **38.7** | 60.5G | 22.8M | 40.3 |
| Mask2Former | R50 | 42.9 | 79.8 | 47.6 | 35.6 | 58.0 | 42.1 | 153G | 45.2M | 26.6 |
| MaskFormer | R50 | 37.4 | 77.3 | 41.2 | 31.8 | **58.8** | 40.0 | 143.0G | 45.2M | 34.3 |
| kMaX-DeepLab | R50 | 36.9 | 75.7 | 42.7 | 28.4 | 20.1 | 26.4 | 280.0G | 51.4M | 14.9 |
| YOSO | R50 | 37.3 | 76.9 | 42.7 | 29.2 | 49.2 | 40.7 | 119.0G | 45.1M | 36.3 |
| **RMP-SAM(Ours)** | R50 | **46.9** | **80.8** | **51.6** | **39.8** | 57.9 | **46.2** | 123.0G | 47.2M | 35.1 |
| MaskFormer | TopFormer | 31.6 | 75.9 | 33.5 | 28.6 | **56.1** | 38.2 | 60.0G | 12.8M | 29.9 |
| YOSO | TopFormer | 31.0 | 75.9 | 33.0 | 26.9 | 45.1 | 27.9 | 36.8G | 12.9M | 31.1 |
| **RMP-SAM(Ours)** | TopFormer | **34.6** | **77.0** | **37.8** | **29.8** | 53.3 | **41.7** | 40.1G | 15.0M | 30.7 |

the VIS task, we use mAP as our primary metric. We further report the semantic segmentation results (mIoU) and video panoptic segmentation results (STQ (Weber et al., 2021) and VPQ (Kim et al., 2020)). Speed testing is conducted fairly on one A100 GPU for all models.

**Re-implemented Baselines.** Since no previous works explore joint co-training for image, video, and interactive segmentation in one model. We re-implement several representative baselines using the same codebase, including non-real-time models (Mask2Former (Cheng et al., 2022), kMaX-DeepLab (Yu et al., 2022)) for reference. We benchmark multiple different methods, including different backbones and decoders.

**Implementation Details.** We implement our models and all other baselines in PyTorch (Paszke et al., 2019). We use the distributed training framework with 16 A100 GPUs. Each mini-batch has two images per GPU, and each batch contains one data type. In particular, we adopt pseudo video training on COCO by moving image masks with random directions. All the models are trained with 12 epochs. For data augmentation, we adopt large-scale jitter as previous works (Cheng et al., 2022) to build strong baselines. For all models, we adopt the same training steps and optimizers. Refer to the appendix (Sec. 7.1) for more details.

## 4.1 MAIN RESULTS

**Our Benchmark Results.** We list our benchmark results in Tab. 3. We benchmark recent state-of-the-art methods using the same training and test settings. From the table, our proposed RMP-SAM achieves the best speed and accuracy trade-off on all three visual segmentation tasks under the various backbones. Mask2Former (Cheng et al., 2022) achieves similar or partially stronger results than our method. However, their speed is still limited, and they are 7-8 FPS slower than our methods. YOSO (Hu et al., 2023) runs as fast as our method. However, the performance gap is still significant. Thus, our proposed RMP-SAM is a simple yet effective baseline for real-time and Multi-Purpose settings.

**Compare with SAM-like Methods.** Tab. 4 provides a detailed comparison between our models and previous SAM-like models. We report the results of mask AP of the COCO validation set, the parameter count, and FLOPs of each model. We first generate bounding boxes for each image using Mask R-CNN with ResNet50. Then, we send these bounding boxes to SAM-like models as box prompts. Our method is co-trained using SAM data and demonstrates even better results than most previous efficient SAM-like methods. It has strength in computational efficiency, with an additional function for video instance segmentation and panoptic segmentation.

**Comparison with Specific Design Models on VIP-Seg.** In Tab. 5(a), we verify the effectiveness of RMP-SAM on a more challenging video segmentation task, video panoptic segmentation (VPS) on the VIP-Seg dataset. Compared with recent works (Li et al., 2023e; Miao et al., 2022), RMP-SAM also archives the best speed and accuracy trade-off, despite RMP-SAM is not specifically designed for VPS.

**Comparison with Specific Design Models on ADE-20k.** In Tab. 5(b), we further transfer RMP-SAM on ADE-20k datasets. Since our method is a real-time model, we only list several representative works for reference. Our method still achieves stronger results than recent work (Hu et al., 2023).

Table 4: Comparison with SAM-like methods on COCO instance segmentation dataset. We first generate bounding boxes using Faster R-CNN, which are used as visual prompts for segmentation. We report the results of mask AP of the COCO validation set, the parameters, and the FLOPs.

| Method | mAP | #Param. | FLOPs(G) | PS | VIS | Interactive |
|---|---|---|---|---|---|---|
| SAM-H (Kirillov et al., 2023) | 35.6 | 641M | 3000 | ✗ | ✗ | ✗ |
| TinySAM (Shu et al., 2023) | 32.5 | 10M | 42 | ✗ | ✗ | ✗ |
| MobileSAM (Zhang et al., 2023a) | 32.1 | 10M | 42 | ✗ | ✗ | ✗ |
| EdgeSAM (Zhou et al., 2023) | 31.9 | 10M | 22 | ✗ | ✗ | ✗ |
| EfficientSAM (Xiong et al., 2023) | 32.4 | 10M | - | ✗ | ✗ | ✗ |
| FastSAM (Zhao et al., 2023) | 34.3 | 68M | 344 | ✗ | ✗ | ✗ |
| RMP-SAM | 33.2 | 47M | 35 | ✓ | ✓ | ✓ |

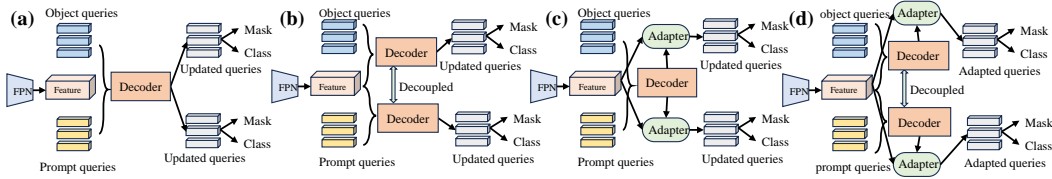

Figure 3: Meta-architecture exploration. (a), Simple shared decoder design. (b), Decoupled decoder design with two heads. (c), Shared decoder with decoupled adapter. (d), Decoupled decoder with the decoupled adapters. Best viewed in color and zoom in.

Table 5: Compared with specifically designed segmentation models.

(a) Comparison with video segmentation method on VIP-Seg validation set.

| Method | backbone | VPQ | STQ | FPS |
|---|---|---|---|---|
| Clip-PanoFCN | R50 | 22.9 | 31.5 | 8 |
| Video K-Net | R50 | 26.1 | 33.1 | 10 |
| Tube-Link | STDCv1 | 30.6 | 32.0 | 14 |
| Tube-Link | STDCv2 | 31.4 | 32.8 | 12 |
| Ours | R18 | 32.5 | 33.7 | 30 |

(b) Comparison with panoptic segmentation method on ADE-20k validation set.

| Method | backbone | PQ | Parameters | FLOPs |
|---|---|---|---|---|
| PanSegFormer | R50 | 36.4 | 51M | 214G |
| MaskFormer | R50 | 34.7 | 45.2M | 181G |
| Mask2Former | R50 | 39.7 | 45.2M | 226G |
| YOSO | R50 | 38.0 | 45.1M | 176G |
| RMP-SAM | R50 | 38.3 | 47.2M | 179G |

## 4.2 ABLATION STUDY AND VISUAL ANALYSIS

**Ablation on Meta-Architecture Design.** In Tab. 6(a), we further explore the meta-architecture design, as shown in Fig. 3. We use R50 as the backbone and keep the adapter unchanged. The table shows that a shared decoder architecture achieves the best parameter and performance trade-off. These findings may conflict with recent multi-task work (Li et al., 2022c) since the decoupled decoders may have more capacity. We argue that since our feature extractors are weaker than those works and the representation power is far from the heavier backbone such as Swin (Liu et al., 2021) or ViT-large (Dosovitskiy et al., 2021), adding more capacity in the decoder may not boost the final performance.

**Effectiveness of Joint Co-training.** We further explore the impact of each dataset in Tab. 6(b). We find joint co-training with image and video data leads to better performance for video instance segmentation but reduces the performance of panoptic segmentation. Adding COCO-SAM training leads to a few performance drops. This is because the learning targets are different for object queries and prompt queries.

**Ablation on Shared Decoder Design.** Tab. 6(c) shows using simple pooling-based dynamic convolution performs well under real-time settings, where we adopt ResNet18 as the backbone and keep the adapter unchanged. In particular, there is no difference in panoptic segmentation results and a slight gap for interactive segmentation. However, per-pixel cross-attention introduces extra computation costs.

**Ablation on Adapter Design.** Then, we explore the adapter design in Tab. 6(d). In particular, we use a pre-trained model without adapters for initialization to see the effect of different adapters.

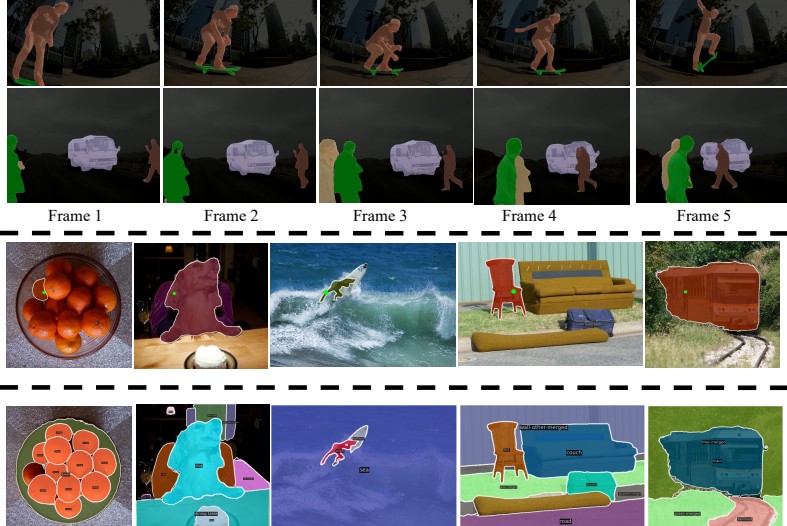

| | | | | |
|---|---|---|---|---|
| Frame 1 | Frame 2 | Frame 3 | Frame 4 | Frame 5 |

Figure 4: The visualization results of YouTube-VIS 2019 and COCO datasets. The first two rows visualize five frames of inputs. The same instances are in the same color. The third row shows the interactive segmentation results with a single-point prompt (green color). The last row shows the panoptic segmentation results.

Table 6: Ablation study on RMP-SAM's component design.

(a) Ablation on meta-architecture.

| Setting | COCO-Panoptic | COCO-SAM | Parameters |
|---|---|---|---|
| Fig. 3 (a) | 43.8 | 54.2 | 46.3M |
| Fig. 3 (b) | 44.0 | 55.2 | 53.6M |
| Fig. 3 (c) | 44.6 | 56.7 | 47.3M |
| Fig. 3 (d) | 45.2 | 56.2 | 54.6M |

(b) Ablation on joint co-training. a: COCO-Panoptic. b : YT-VIS 2019. c: COCO-SAM.

| Setting | COCO-Panoptic | YT-VIS 2019 | COCO-SAM |
|---|---|---|---|
| a | 36.6 | - | - |
| b | - | 21.5 | - |
| a + b | 36.2 | 36.0 | 50.7 |
| a + b + c | 35.7 | 35.3 | 50.9 |

(c) Ablation on shared decoder design. DC: dynamic convolution. DCG: dynamic convolution with gating.

| Settings | COCO-Panoptic | COCO-SAM |
|---|---|---|
| Per-Pixel Cross-Attention | 45.0 | 55.3 |
| Pooling + DC | 43.8 | 55.7 |
| Pooling + DCG | 44.6 | 56.7 |

(d) Ablation on adapter design. CA: cross-attention. DC: dynamic convolution.

| $A_{obj}$ | $A_{prompt}$ | COCO-Panoptic | COCO-SAM |
|---|---|---|---|
| - | - | 44.2 | 53.2 |
| CA | CA | 42.6 | 54.3 |
| DC | DC | 44.7 | 52.1 |
| DC | CA | 44.6 | 56.7 |
| CA | DC | 44.5 | 56.2 |

In particular, we find that using asymmetric adapters works well for balanced results for object queries and prompt queries since the goals of the two queries are different. The former needs the spatio-temporal and scene-level context, while the latter only cares about the region context with the input position as the guidance. Our adapter can also work with other architecture, as shown in the appendix (Sec. 7.2).

**Result Visualization.** In Fig. 4, we give the three different visualization results on the YouTube-VIS 2019 and COCO datasets. We generate these visualization results using only a single model thanks to the joint co-training strategy. The first three rows in Fig. 4 show the video instance segmentation. The fourth row shows the interactive segmentation results with a single-point prompt (shown in green color), and the last row shows the results of panoptic segmentation.

# 5 RELATED WORK

**Universal Segmentation.** Prior research has focused on designing segmentation models tailored for specific tasks. Recent developments in this field (Dosovitskiy et al., 2021; Liu et al., 2021; Carion et al., 2020; Zhang et al., 2021; Cheng et al., 2022; Yu et al., 2022; Li et al., 2021b; 2023c; Gu et al., 2023; Yuan et al., 2022; Li et al., 2022c; 2023d; 2024b; Yuan et al., 2024; Yang et al., 2021b;

2024) have shifted towards employing mask classification architectures, coupled with an end-to-end set prediction objective, to facilitate universal segmentation. This approach has outperformed specialized models (Chen et al., 2019; Kirillov et al., 2019; He et al., 2017; Li et al., 2021a; Huang et al., 2019; Yuan et al., 2020; Zhao et al., 2017; Chen et al., 2018; Li et al., 2022b; 2020a; Chen et al., 2017) across various segmentation tasks, including those involving images, videos, and point clouds (Kim et al., 2022; Li et al., 2022d; 2023e; Cheng et al., 2021a; Xu et al., 2022). Specifically, Mask2Former (Cheng et al., 2022) uses a masked-attention mechanism and surpasses the performance of prior specific image segmentation models. Similarly, Tube-Link (Li et al., 2023e) outperforms earlier models (Li et al., 2022d;b; Liu et al., 2020; Shin et al., 2023; Goel et al., 2021) tailored for specific video segmentation tasks. Semantic-SAM (Li et al., 2023a), built on Mask-DINO (Li et al., 2023b), employs a hybrid query design to facilitate panoptic and interactive segmentation within a single framework. However, despite the strong performance of these works, their complexity, stemming from the heavier encoder or the cascaded decoder designs, poses significant challenges for integration into actual products. Slow processing speeds and substantial parameter overhead primarily hinder these.

**Efficient Model Design.** This research direction primarily concentrates on the development of efficient CNNs (Howard et al., 2017; Sandler et al., 2018; Howard et al., 2019; Iandola et al., 2016; Zhang et al., 2018; Ma et al., 2018; Han et al., 2020), transformers (Maaz et al., 2022; Pan et al., 2022; Mehta & Rastegari, 2022), and hybrid architectures (Zhang et al., 2023b; Ma et al., 2022; Chen et al., 2022b; Li et al., 2022a; Zhou et al., 2022), aimed at advancing visual representation learning. Most research investigates backbone design within the constraints of real-time processing and parameter efficiency. These studies are orthogonal to our work, which primarily explores the impact of data, task, and decoder design. Additionally, our work employs efficient models as encoders and presents detailed benchmarks and analyses of the encoder's effects.

**Efficient Segmentation.** Previous studies (Zhao et al., 2018; Li et al., 2020b; Yu et al., 2018; Hu et al., 2023; Hong et al., 2021; Wan et al., 2023; Yu et al., 2021; Mehta et al., 2018) on efficient segmentation have predominantly concentrated on closed-set and specific domains. In particular, a significant portion of this research (Li et al., 2020b; Hu et al., 2023; Mehta et al., 2019) is dedicated to driving scenarios. Recently, various studies (Zhang et al., 2022; Wan et al., 2023; Zhou et al., 2023) have developed efficient segmentation techniques that facilitate model execution on mobile devices. Mobile SAM (Zhang et al., 2023a) introduces a streamlined encoder distillation method. Fast SAM (Zhao et al., 2023) employs a single-stage instance segmentation framework that directly decodes class-agnostic masks. Recently, multiple studies have been conducted on efficient panoptic segmentation (Sun et al., 2023; Hu et al., 2023; Mohan & Valada, 2021) and rapid video instance segmentation (Yang et al., 2021a; Zhang et al., 2024). However, these real-time segmentation methods are limited to specific tasks. We contend that a multi-purpose model capable of real-time performance would have wide-ranging applications in editing, tracking, and segmentation functionalities within various products.

**Interactive Segmentation.** Interactive segmentation (Kirillov et al., 2023; Chen et al., 2022a; Sofiiuk et al., 2022; Lin et al., 2020; 2022) distinguishes objects by actively incorporating user inputs. Recently, SAM (Kirillov et al., 2023) has shown remarkable segmentation outcomes by leveraging multiple user interaction inputs and training on SA-1B data. Inspired by SAM, several studies (Zhang et al., 2023a; Zhou et al., 2023) have concentrated on employing a lighter image encoder to adapt SAM for real-time scenarios.

## 6 CONCLUSION

In this work, we explore a new challenging setting, real-time multi-purpose segmentation, to explore multi-task segmentation in one efficient model. To achieve this, we first benchmark existing image segmentation approaches by extending their query for video and interactive segmentation. However, no work can achieve the best speed and accuracy trade-off on all three sub-tasks. To solve this problem, we introduce RMP-SAM, a real-time segmentation model. It has a simple shared decoder with two key designs: shared dynamic convolution and asymmetric adapters. These designs lead to the best trade-off between speed and accuracy on three segmentation tasks. The proposed RMP-SAM model is versatile and effective for various tasks, including image segmentation, video segmentation, and iterative segmentation. To our knowledge, no real-time model can achieve this goal. The proposed benchmark and RMP-SAM baseline can facilitate future research in this new setting.

## ACKNOWLEDGMENTS

This work is supported by the National Key Research and Development Program of China (No.2023YFC3807603). This work is also supported by the Intelligence Advanced Research Projects Activity (IARPA) via Department of Interior/ Interior Business Center (DOI/IBC) contract number 140D0423C0074. The U.S. Government is authorized to reproduce and distribute reprints for Governmental purposes notwithstanding any copyright annotation thereon. Disclaimer: The views and conclusions contained herein are those of the authors and should not be interpreted as necessarily representing the official policies or endorsements, either expressed or implied, of IARPA, DOI/IBC, or the U.S. Government.

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

## 7 APPENDIX

In this appendix, we add a deeper discussion on our RMP-SAM with existing unified methods from different views in Sec. 7.1. Next, we present more detailed results to support the benchmark and our proposed RMP-SAM in Sec. 7.2. Then, in Sec. 7.3, we present a more detailed visual comparison using our model. Finally, in Sec. 7.4, we discuss several failure cases and potential future work. Moreover, we have appended a short video to introduce our work briefly and append the code for reference.

### 7.1 MORE METHOD DETAILS

**More Implementation Details.** We implement our proposed RMP-SAM and benchmark methods using PyTorch and train them in the same environment. For kMaX-DeepLab (Yu et al., 2022), we excluded the auxiliary semantic segmentation loss, instance discrimination loss, and preserving PQ-style loss alignment with Mask2former (Cheng et al., 2022) for a fair comparison. We utilize the AdamW (Loshchilov & Hutter, 2019) optimizer with a weight decay of 0.05, and the learning rate is set to 1e-4 for all methods. We warm up the learning rate in the first 500 iterations using a linearly increased strategy and decay the learning rate at 8 and 11 epochs by a factor of 10. For data augmentation, we use the large-scale jittering (LSJ) augmentation (Ghiasi et al., 2020) with a random scale sampled from the range of 0.1 to 2.0 and followed by a fixed size crop to 1024×1024. For joint dataset training, we sample clip images with a ratio of 1: 25: 1 for COCO-panoptic, Youtube-2019, and SAM datasets. For a fair comparison with efficient SAM models, we adopt the SAM dataset for fine-tuning. In particular, we adopt the co-trained checkpoint and fine our visual prompt segmentation branch using 10% SAM dataset. The training follows the previous works (Zhang et al., 2023a).

**More Inference Details.** For video instance segmentation, we adopt simple object query matching (Li et al., 2022d; Huang et al., 2022) for all methods to keep the fair comparison to assign instance ID. We mainly use point prompts for interactive segmentation, which are more challenging than box prompts. For panoptic segmentation, all hyperparameters are the same. The FPS is obtained on one 40GB A100 GPU for the main paper and supplementary.

**Detailed Comparison with SAM-2.** We present a more detailed comparison with SAM-2 (Ravi et al., 2024) here. (1) For functionality, SAM-2 mainly focuses on video object segmentation and interactive segmentation. Compared with SAM, it adds mask tracking ability on given visual prompts, while our method mainly explores multi-purpose and multi-task real-time segmentation. We unify panoptic segmentation, video instance segmentation, and interactive segmentation in one model with the requirements in real time. (2) For data scale and diversity, SAM-2 builds a large scale dataset and mainly has one purpose: Video Object Segmentation. Our RMP-SAM only involves a small set of public data sources and has multiple purposes: segmentation, mask labeling, mask tracking, and panoptic segmentation. (3) For goals, SAM-2 aims at the production level with large-scale datasets (including internal datasets) co-training. Our RMP-SAM aims at efficient model design and performs well under real-time constraints. (4) Last, our work is concurrent with SAM-2 since our work is also inspired by pioneering SAM.

### 7.2 MORE EXPERIMENT RESULTS

**Benchmark More Methods.** In Tab. 7, we benchmark more models and backbones (Zhang et al., 2021; Maaz et al., 2022; Mehta & Rastegari, 2022) as the supplementary to main paper. Again, for various backbones and different heads, our model can achieve better or comparable results across three different tasks while running in real-time. In particular, compared with the dynamic kernel-based method, K-Net, our approach achieves better performance on the R18 and R50 backbones and comparable results on the lightweight backbones (EdgeNexT).

**Scale Up Joint Co-training.** In the bottom of Tab. 7, we further scale up our model on a large backbone, ConvNeXt large (Liu et al., 2022). From the table, we find there are significant gains in all settings while still running with 8 FPS. This means our proposed multi-purposed segmentation still has room to balance the accuracy and inference speed when adopting the large model. This also verifies the scalability of RMP-SAM.

| Method | Backbone | COCO-Panoptic | | | | COCO-SAM mIoU | YouTube-VIS 2019 mAP | FLOPs | Parameters | FPS |
|---|---|---|---|---|---|---|---|---|---|---|
| | | PQ | SQ | PQ$_{th}$ | PQ$_{st}$ | | | | | |
| YOSO | EdgeNexT | 31.8 | 74.5 | 35.6 | 26.1 | 47.4 | 35.4 | 40G | 15.3M | 28.6 |
| K-Net | R18 | 33.1 | 75.7 | 36.8 | 27.4 | 53.6 | 34.8 | 124G | 25.6M | 30.4 |
| K-Net | R50 | 39.2 | 78.2 | 43.9 | 31.6 | 56.3 | 41.7 | 171G | 38.5M | 20.9 |
| K-Net | EdgeNexT | 38.0 | 76.7 | 42.5 | 31.4 | 55.7 | 44.0 | 108G | 19.5M | 30.5 |
| Mask2Former | EdgeNexT | 39.8 | 78.8 | 43.6 | 34.0 | 54.2 | - | 73G | 12.0M | 25.6 |
| RMP-SAM | R18 | 39.9 | 78.6 | 43.3 | 34.8 | 52.7 | 38.7 | 60.5G | 22.8M | 40.3 |
| RMP-SAM | R50 | 46.9 | 80.8 | 51.6 | 39.8 | 57.9 | 46.2 | 123.0G | 47.2M | 35.1 |
| RMP-SAM | EdgeNexT | 38.0 | 77.7 | 42.1 | 31.9 | 54.8 | 46.3 | 44G | 14.3M | 31.5 |
| RMP-SAM | ConvNeXt-L | 52.3 | 82.5 | 58.6 | 42.8 | 61.1 | 55.6 | 700G | 239M | 8.6 |

Table 7: Benchmark More Models on Real-Time multi-purpose Segmentation. The GFLOPs are obtained with $1333 \times 800$ inputs.

Table 8: Explore adapter Design on More Methods. We apply our dual adapter on the Mask2Former architecture and find it works well.

| Method | Backbone | PQ | mIoU |
|---|---|---|---|
| Mask2Former | R50 | 43.2 | 57.0 |
| Mask2Former + adapter | R50 | 43.0 | 58.1 |

Table 9: Compared with real-time click-based interactive segmentation model.

| Method | 1-IoU | FPS | FLOPs |
|---|---|---|---|
| SAM(L) | 55.7 | 1.2 | 1315.0 |
| Semantic-SAM(L) | 57.0 | 2.3 | 745.8G |
| RMP-SAM (ours) | 57.9 | 35.1 | 123.0G |

**Explore Dual Adapter Design on Other Methods.** We explore the effectiveness of our dual adapter on Mask2Former (Cheng et al., 2022) and add the adapter module to the last decoder layer of Mask2Former. As shown in Tab. 8, Mask2Former with adapter will improve 1.1 mIoU score compared with origin Mask2Former for interactive segmentation and only drop 0.3 PQ for panoptic segmentation. This result indicates that the adapter module applies to our method and can be generalized to other methods.

**Single-granularity Interactive Segmentation.** In Tab. 9, we evaluate the 1-click mIoU (denoted as 1-IoU) for SAM (Kirillov et al., 2023), Semantic-SAM (Li et al., 2023a) and our model on COCO Val2017. Our model surpasses SAM and Semantic-SAM in 1-IoU performance while also being faster and requiring less computation.

**Comparison with Recent Efficient SAM Models.** As shown in Tab. 4, we compare our method with SAM-like methods for coco instance segmentation. In Tab. 10, Tab. 11, Tab. 12. Our method can achieve the stronger results than these counterparts, indicating the effectiveness of model design. We also provide visualization in Fig. 5 for easier and direct comparison.

**Comparison with Strong Video Segmentation Models.** Several works explore stronger video segmentation designs. We argue the scope of our research is orthogonality to the works. Firstly, since our method is not designed for specific models and our method is trained on multiple purposes tasks, directly comparing our method with previous works (Li et al., 2023e; Zhang et al., 2023c) may not be fair. As our core goal is unifying multiple tasks in real-time scenarios and adopting a multi-task co-training strategy, our method may not perform the best on a single task. Secondly, The work (Li et al., 2024a) also designs a unified model for video segmentation tasks. However, it cannot perform SAM-like segmentation in real time. We adopt joint image-video data co-training rather on image or video data. Thus, we can keep SAM-like segmentation, panoptic segmentation, and video instance segmentation in one model and run in real time. Thirdly, all these methods focus on achieving strong results rather than real-time design. The work (Zhang et al., 2023c) uses extra transformer encoders after heavy Mask2Former architectures, bringing more computation costs. We follow the work (Li et al., 2023e; Zhang et al., 2023c), with the modification on replacing pre-training on COCO with our co-training on COCO, COCO-SAM, and YTVIS-2019. Then, we follow the works (Li et al., 2023e; Zhang et al., 2023c) and finetune the pre-trained model on YTVIS-2019. As shown in Tab. 14, nevertheless, we achieved an mAP of 47.2 when using the ResNet-50 backbone. To provide more

Table 10: Compared with SAM-like methods on COCO instance segmentation. We adopt the Mask R-CNN with ResNet50 as a detector. The bounding box obtained from Mask R-CNN is adopted as a box prompt for SAM-like methods. We report average precision at various thresholds. The $AP_s$, $AP_m$, and $AP_l$ mean the mean average precision for small, medium, and large objects.

| Method | mAP | $AP_{50}$ | $AP_{75}$ | $AP_s$ | $AP_m$ | $AP_l$ | #param. | FLOPs(G) |
|---|---|---|---|---|---|---|---|---|
| SAM-H | 35.6 | 54.9 | 38.4 | 17.2 | 39.1 | 51.4 | 641M | 3000 |
| TinySAM | 32.7 | 53.6 | 34.0 | 15.5 | 35.3 | 48.3 | 10M | 42 |
| MobileSAM | 32.3 | 53.0 | 33.6 | 14.7 | 25.3 | 48.6 | 10M | 42 |
| EdgeSAM | 32.2 | 53.4 | 33.3 | 14.8 | 35.0 | 47.0 | 10M | 22 |
| EfficientSAM | 32.6 | 53.3 | 33.9 | 16.2 | 35.2 | 47.1 | 10M | - |
| RMP-SAM(Ours) | 33.5 | 54.6 | 35.4 | 14.3 | 36.3 | 50.8 | 47M | 35 |

Table 11: Compared with SAM-like methods on COCO instance segmentation and adopt Mask R-CNN with ResNet101 as a detector.

| Method | mAP | $AP_{50}$ | $AP_{75}$ | $AP_s$ | $AP_m$ | $AP_l$ | #param. | FLOPs(G) |
|---|---|---|---|---|---|---|---|---|
| SAM-H | 38.4 | 59.4 | 41.3 | 24.1 | 46.9 | 56.9 | 641M | 3000 |
| TinySAM | 35.4 | 57.8 | 36.9 | 17.6 | 39.2 | 51.8 | 10M | 42 |
| MobileSAM | 34.9 | 57.2 | 36.2 | 17.0 | 38.9 | 52.1 | 10M | 42 |
| EdgeSAM | 34.9 | 57.6 | 36.2 | 17.3 | 38.9 | 50.3 | 10M | 22 |
| EfficientSAM | 35.2 | 57.6 | 36.5 | 18.2 | 39.0 | 51.0 | 10M | - |
| RMP-SAM(Ours) | 35.6 | 58.0 | 36.9 | 17.4 | 40.2 | 53.3 | 47M | 35 |

comparison, we use ConvNext-L as our backbone and train a model based on the joint co-training described in the paper, achieving a result of 62.2. The results indicate our method can still achieve comparable results with these expert models.

**Testing the efficiency across different GPU platforms.** We have evaluated several methods from Table 2 across multiple GPU platforms, including A100-40G, A10-22G, and 3090-24G. Unfortunately, we currently do not have access to a V100 GPU and are unable to provide its corresponding results.

All model results for each specific GPU are generated on the same machine. The FPS and GFlops values are calculated using a 1333 x 800 pixels resolution image. We report these results with the ResNet-18 backbone.

## 7.3 VISUAL COMPARISON

**Attention Map Visualization.** As shown in Fig. 6, we generate the heatmap to illustrate our integrated multi-tasks, which include panoptic segmentation, interactive segmentation, and video instance segmentation. We sample four continuous frames from the YouTube-VIS 2019 validation set and visualize a randomly selected object. The heatmap is the attention score between object/prompt query and image features. The results indicate that despite using a shared decoder across different

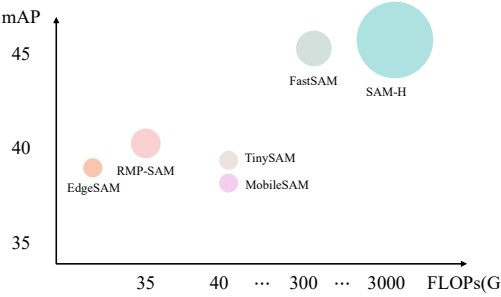

Figure 5: The visualization results of SAM-like methods on COCO instance segmentation. We adopt the ViTDet with ViT-b as the detector. The larger dot means more parameters.

Table 12: Compared with SAM-like methods on COCO instance segmentation and adopt ViTDet with ViT-b as a detector.

| Method | mAP | $AP_{50}$ | $AP_{75}$ | $AP_s$ | $AP_m$ | $AP_l$ | #param. | FLOPs(G) |
|---|---|---|---|---|---|---|---|---|
| SAM-H | 45.7 | 69.4 | 49.8 | 26.9 | 49.0 | 64.1 | 641M | 3000 |
| TinySAM | 39.2 | 64.5 | 40.4 | 23.7 | 42.3 | 55.7 | 10M | 42 |
| MobileSAM | 38.3 | 63.5 | 39.3 | 22.3 | 41.7 | 55.3 | 10M | 42 |
| EdgeSAM | 39.0 | 64.8 | 40.1 | 23.8 | 42.3 | 54.5 | 10M | 22 |
| EfficientSAM | 39.4 | 64.9 | 40.9 | 23.8 | 42.3 | 55.3 | 10M | - |
| RMP-SAM | 41.0 | 65.9 | 42.5 | 24.2 | 44.7 | 57.3 | 47M | 35 |

Table 13: Testing the efficiency across different GPU platforms.

| Method | GPU | FLOPs | Parameters | FPS |
|---|---|---|---|---|
| Mask2Former | A100-40G | 89.8G | 18.6M | 31.2 |
| kMaX-DeepLab | A100-40G | 87.1G | 18.7M | 15.0 |
| YOSO | A100-40G | 57.3G | 18.7M | 41.0 |
| RMP-SAM | A100-40G | 60.5G | 22.8M | 40.3 |
| Mask2Former | A10-22G | 89.8G | 18.6M | 10.1 |
| kMaX-DeepLab | A10-22G | 87.1G | 18.7M | 4.3 |
| YOSO | A10-22G | 57.3G | 18.7M | 13.6 |
| RMP-SAM | A10-22G | 60.5G | 22.8M | 14.2 |
| Mask2Former | 3090-24G | 89.8G | 18.6M | 25.6 |
| kMaX-DeepLab | 3090-24G | 87.1G | 18.7M | 9.0 |
| YOSO | 3090-24G | 57.3G | 18.7M | 31.4 |
| RMP-SAM | 3090-24G | 60.5G | 22.8M | 32.0 |

Table 14: Comparison with stronger video instance segmentation methods. Our RMP-SAM can achieve comparable results with more functionalities on image panoptic segmentation and interactive segmentation.

| Method | Backbone | Youtube-VIS 2019 | COCO-SAM | COCO-Panoptic |
|---|---|---|---|---|
| UniVS | R50 | 47.4 | - | - |
| DVIS | R50 | 51.2 | - | - |
| Tube-Link | R50 | 52.8 | - | - |
| RMP-SAM | R50 | 47.2 | - | - |
| UniVS | Swin-L | 60.0 | - | - |
| DVIS | Swin-L | 63.9 | - | - |
| Tube-Link | Swin-L | 64.6 | - | - |
| RMP-SAM | ConvNeXt-L | 62.2 | 60.8 | 52.0 |

tasks, each task can effectively learn. This indicates that our design is effective and can significantly reduce the computational complexity of the model.

**Comparison on Youtube-VIS dataset.** In Fig. 7, we compare our RMP-SAM (right) with a strong baseline Mask2Former (left) with the same ResNet50 backbone. Our methods achieve more consistent tracking and segmentation results in these examples.

**Comparison on COCO Panoptic Segmentation dataset.** In Fig. 8, We demonstrate that our RMP-SAM model surpasses K-Net in panoptic segmentation tasks. RMP-SAM exhibits enhanced accuracy in processing complex scenes and maintaining fine details, especially in segmenting object edges and managing overlapping objects.

**More Interactive Segmentation Results.** In Fig. 9, we present further visualizations of interactive segmentation, showcasing segmentation outcomes for 'things' and 'stuff'. In these visualizations, green dots signify manually specified prompts. Our model exhibits precise segmentation abilities

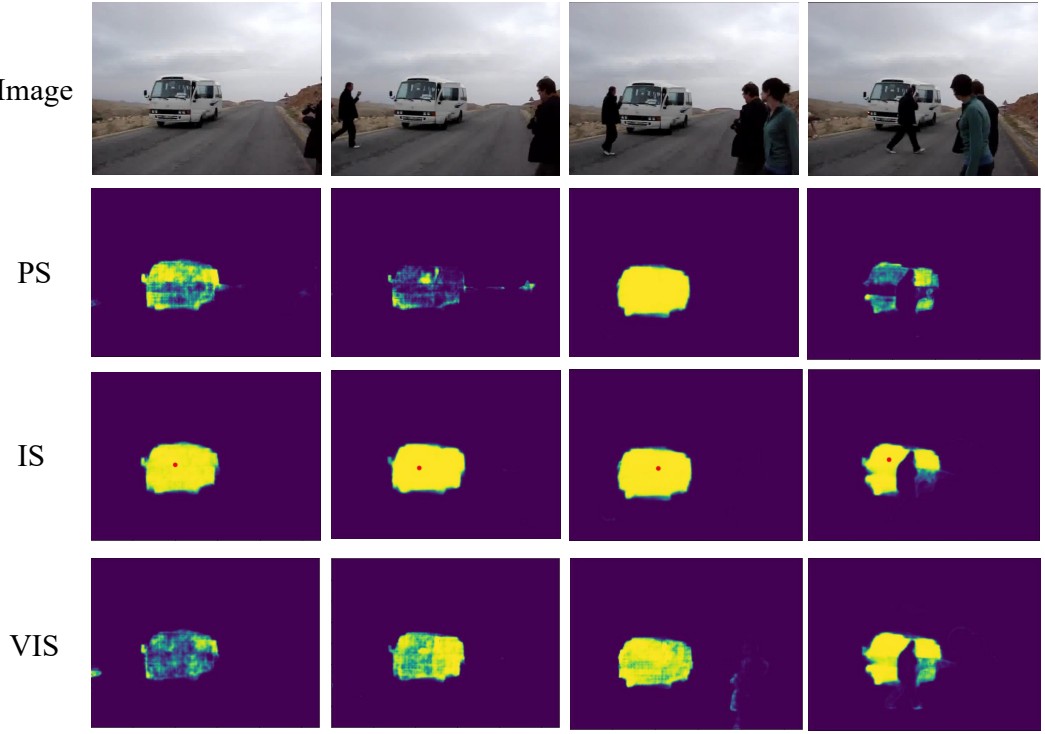

Figure 6: The heatmap shows the results of three distinct tasks: panoptic, interactive, and video instance segmentation, represented by PS, IS, and VIS, respectively. The four test images are sampled from the YouTube-VIS 2019 validation set, and the heatmap is the attention score between object/prompt query and image features. The red point in IS is the point prompt.

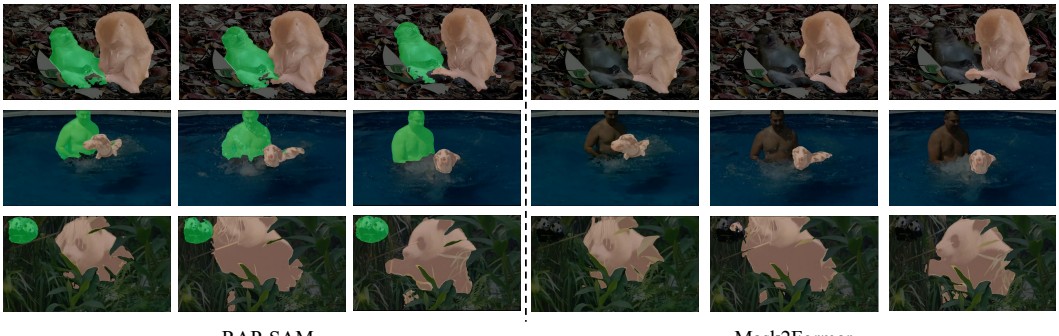

RAP-SAM                                        Mask2Former

Figure 7: When compared to the Mask2Former(right) on the YouTube-VIS 2019 dataset, our RMP-SAM (left) demonstrates superior performance in recognizing and segmenting objects in certain scenarios, and it also holds an advantage in detailing edges.

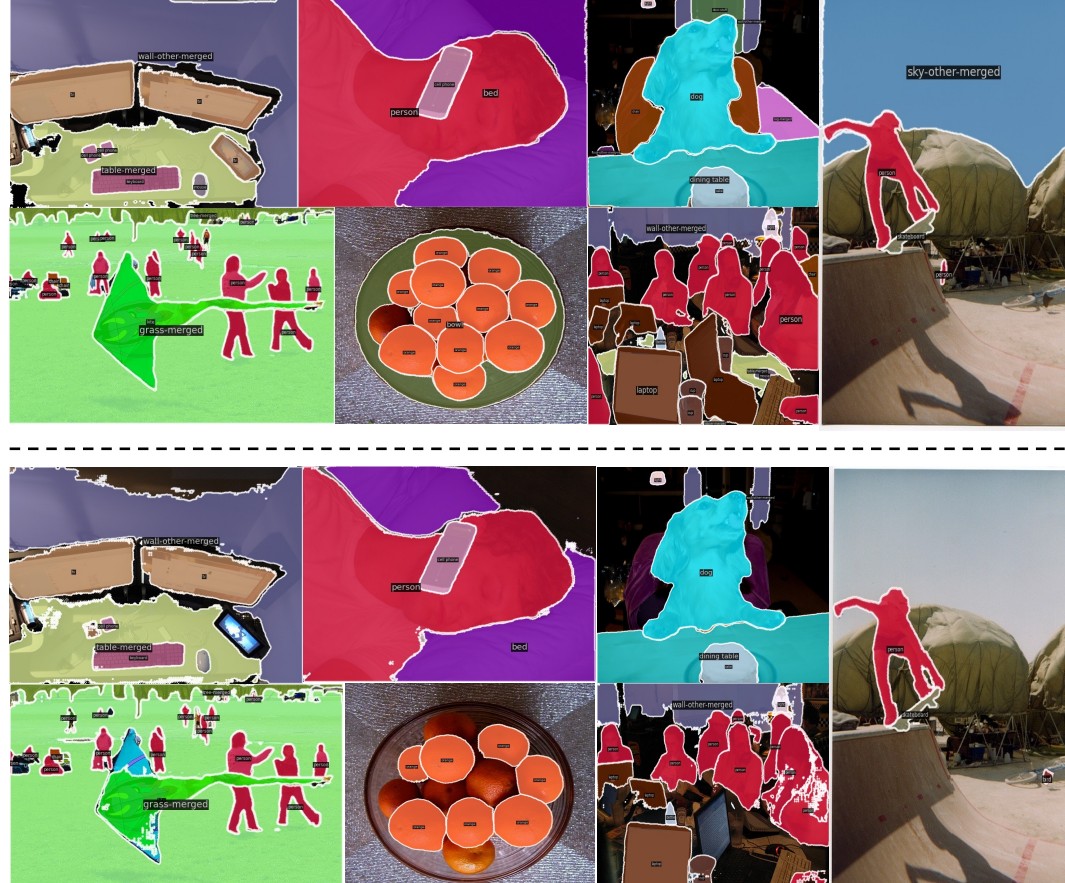

Figure 8: When compared with K-Net (bottom) on the COCO dataset, our RMP-SAM (top) shows a significant advantage, achieving more accurate segmentation of foreground and background, with smoother edges.

Things

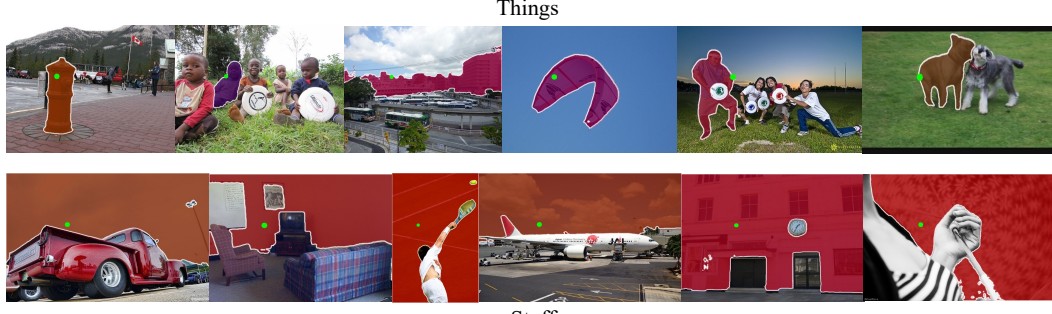

Stuff

Figure 9: Here are additional results of interactive segmentation. For point prompts or box prompts, RMP-SAM segments out the corresponding object based on the spatial location of the prompt. We have provided both segmentation results for 'things' and 'stuff', demonstrating that RMP-SAM can segment both while maintaining strong performance.

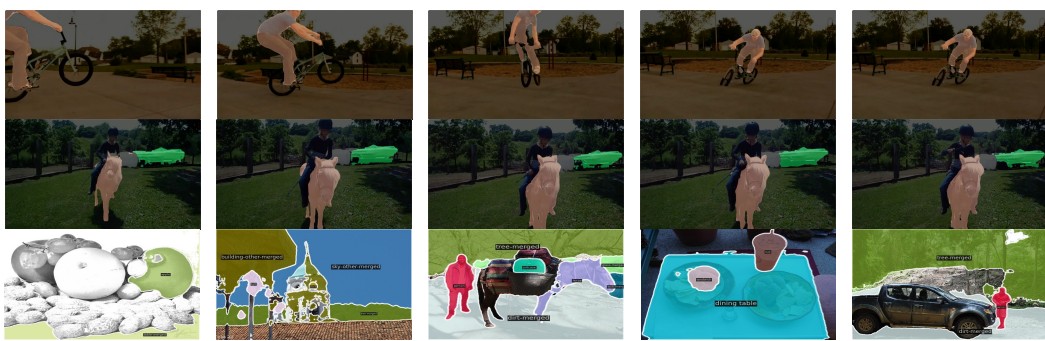

Figure 10: Failure modes of RMP-SAM in image/video segmentation.

for 'things' and 'stuff'. Notably, our model accurately identifies and segments the object even when prompt points are situated on object boundaries.

### 7.4 LIMITATIONS AND FUTURE WORK

**Failure Cases Analysis.** We also analyze failure cases in video/image segmentation and identify two typical failure modes of RMP-SAM. Firstly, as illustrated in Fig. 10, when faced with multiple objects with high overlap, RMP-SAM struggles to fully recognize them, resulting in the omission of several masks. Secondly, in crowded scenarios, RMP-SAM faces challenges on recognizing and segmenting all instances with a limited number of instance kernels.

**Future Work Discussion.** There are several directions to explore for real-time multi-purpose segmentation. The first direction is to achieve better-balanced performance on image, video, and interactive segmentation (since there is still a lot of room for performance, as shown in the last row of Tab. 7). The second direction is to speed up the model and deploy the model on the edge device, such as the iPhone. The third direction is to explore different knowledge distillation approaches to transfer the vision foundation model in real-time multi-purpose models. These will be our future work.

