# OpenReview forum: "RMP-SAM: Towards Real-Time Multi-Purpose Segment Anything"
_ICLR.cc/2025/Conference — ICLR 2025 Oral_

### Official Review · Reviewer_HZhq · 2024-10-21

**Soundness:** 3
**Presentation:** 2
**Contribution:** 3
**Rating:** 8
**Confidence:** 3

**Summary:**

The authors explore a novel real-time segmentation setting called real-time multi-purpose segmentation. It contains three fundamental sub-tasks: interactive segmentation, panoptic segmentation, and video instance segmentation. In contrast to previous methods that use a separate design for each task, the authors use only a single end-to-end model to handle all these tasks in real time. To fulfill the real-time requirements and balance multitask learning, a new dynamic convolution-based method, Real-Time Multi-Purpose SAM (RMP-SAM), is introduced. They benchmark several strong baselines by extending existing work to support multi-purpose segmentation.

**Strengths:**

- Large models can perform many tasks, but are not real-time capable because of the large encoders, while real-time models are often specialized in only one task. The method presented here aims to combine the two things, i.e., "the first real-time multi-purpose segmentation model".
- Precise implementation details are given and the comparisons with the other methods appear to be fair.
- The method achieves good results in the trade-off between performance and speed across the various tasks and datasets.
- The ablation studies are useful and show interesting insights.

**Weaknesses:**

- Many architectural elements were adopted from other works, it is not clear to me if there are already similar architectures as proposed here, or where exactly is the innovation (except the jointly training).
- In the related work section, many works are cited and also compared at the task level, but I also miss a comparison at the architectural level.
- The tables, especially table 3, are difficult to read because nothing is in bold print and you have to search for the trade-off here. A plot like Fig. 1b would be more useful.
- The references to the appendix could be a little more precise and there is no reference to Table 2.

**Questions:**

- What does the dot size in Fig. 1b indicate?
- The abstract says "generalization ability of these models across diverse scenarios", a learnable classifier with CLIP text embeddings is also used and “segment anything” is in the title. Is there a connection to open-vocabulary?

---

> ### Author Response · Authors · 2024-11-22
> **Response to Reviewer HZhq**
>
> #### Q1: Many architectural elements were adopted from other works, it is not clear to me if there are already similar architectures as proposed here, or where exactly is the innovation (except the jointly training).
>
> Firstly, we integrate common components from K-Net and YOSO into our network architecture. However, we propose utilizing shared decoders and independent adapters for multi-task co-training within these structures. Our main contribution is the introduction of a new benchmark designed to support various segmentation tasks through a single real-time model. To evaluate this benchmark, we extended several existing real-time segmentation models to handle panoptic, video instance, and interactive segmentation within a unified framework. Finally, we present our model, RMP-SAM, a dynamic convolution-based approach with an improved task adaptation design. RMP-SAM achieves an optimal trade-off between performance, task versatility, and speed. We note that expanding the image along the time dimension is merely a specific implementation strategy and not our core contribution.
>
>
>
> #### Q2: Architectural level comparison
>
> Thank you for your reminder. We have provided a more detailed architectural-level comparison in the related work section. Our architecture incorporates common network designs such as YOSO's lite neck and K-Net's cascade decoder. Unlike K-Net, we unify various tasks into mask prediction by applying Hungarian matching between the predicted and ground truth masks. In contrast to the MaskFormer series, which relies on a masked-attention mechanism, we have developed an efficient and lightweight dynamic convolution framework alongside a decoupled adapter for semantic-aware segmentation and visual prompt-aware segmentation. Most importantly, we have successfully unified multiple tasks within the same architecture, which represents our core contribution. Please the ablation parts and appendix on the effectiveness of decoupled adapter.
>
> #### Q3: The tables, especially table 3, are difficult to read because nothing is in bold print and you have to search for the trade-off here. A plot like Fig. 1b would be more useful.
>
> Thank you very much for your reminder. We have modified the structure of the table by bolding some of the results for easier reading and comparison. Please revisit our paper to review these changes.
>
>
> #### Q4: The references to the appendix could be a little more precise and there is no reference to Table 2.
>
> Thank you very much for your correction. We have checked the references in the appendix and added the connection to Table 2.
>
>
> #### Q5: What does the dot size in Fig. 1b indicate?
>
> Thanks for your suggestion, we have made it clear in our refine draft. We adopt different dot sizes as different model parameters (model size). Large dots mean larger mode sizes.
>
>
> #### Q6: The abstract says "generalization ability of these models across diverse scenarios", a learnable classifier with CLIP text embeddings is also used and “segment anything” is in the title. Is there a connection to open-vocabulary?
>
> As you mentioned, our approach does resemble open vocabulary learning. Replacing the learnable classifier with CLIP text embeddings is a standard operation in open vocabulary learning. For our project, utilizing CLIP text embeddings rather than a learnable classifier is essential for unifying multiple tasks. While our model's ability to recognize unseen objects is important, our primary objective is achieving task unification.

---

> > ### Author Response · Authors · 2024-11-25
> > **Please let us know whether all the issues are addressed**
> >
> > Dear reviewer,
> >
> > Thanks for the comments. We have provided more explanations and answers to your questions. Since the deadline for discussion is Nov 26, please let us know whether we have answered all the questions. Please also consider raising the score after all issues are addressed.
> >
> > Thanks,

---

> ### Comment · Reviewer_HZhq · 2024-11-25
>
> Dear authors,
>
> after reading the other reviews and the answers, I decided to raise my score. Thank you for the replies.

---

> > ### Author Response · Authors · 2024-11-25
> > **Thanks**
> >
> > Dear reviewer,
> >
> > Thanks for raising the score. We have merged your comments into the latest version.
> >
> > Best Regards!
> >
> > Authors of RMP-SAM.

---

### Official Review · Reviewer_NMA5 · 2024-11-01

**Soundness:** 3
**Presentation:** 2
**Contribution:** 2
**Rating:** 6
**Confidence:** 4

**Summary:**

This paper presents a real-time multi-purpose segmentation model called RMP-SAM. RMP-SAM handles various tasks such as interactive segmentation, panoptic segmentation, and video instance segmentation using a single model. To balance the accuracy and speed, RMP-SAM utilizes a lightweight encoder and a dynamic convolution-based decoder. RMP-SAM achieves fast inference while maintaining satisfactory performance.

**Strengths:**

- RMP-SAM unifies interactive segmentation, panoptic segmentation, and video instance segmentation within a single model.
- RMP-SAM offers a good trade-off between speed and accuracy.
- Extensive experiments demonstrate the model's effectiveness.

**Weaknesses:**

- The authors do not provide detailed information for joint training. Joint training for multiple tasks can be complex. How do the authors train RMP-SAM for some potential problems, such as avoiding the model being dominated by a single task and performance degradation by conflicts between different tasks?

- This paper ignores some related methods, making it difficult to assess the model's performance relative to existing SOTA approaches. For example, some universal methods[1,2,3] obtain better results than RMP-SAM using ResNet50. The authors should make a comprehensive comparison with other methods.

[1] Tube-Link: A flexible cross tube framework for universal video segmentation. CVPR 2023.

[2] Dvis: Decoupled video instance segmentation framework. CVPR 2023.

[3] Univs: Unified and universal video segmentation with prompts as queries. CVPR 2024.

**Questions:**

Please see above.

---

> ### Author Response · Authors · 2024-11-22
> **Response to Reviewer NMA5**
>
> #### Q1: Detailed information about joint training.
>
> Thanks for your questions. We agree that joint co-training can be complex and can lead to performance degradation compared with single tasks. We answer your questions in two aspects: technical parts and training data parts:
>
> **Technical parts:** We first unify multiple segmentation tasks for joint co-training under a single paradigm. All training targets are represented as one entity label and mask for all three cases, where the entity can be a thing, stuff, class-agnostic mask, or their corresponding labels. We unify label taxonomies across datasets and replace the learnable classifier with CLIP text embeddings. Hungarian matching is then applied between the predicted and ground-truth entity masks to assign object queries to video/image entities.
> In addition, we present a decoupled decoder design to better balance the semantic-aware masks (panoptic segmentation and video instance segmentation) and prompt-aware masks (interactive segmentation). Please see the ablation parts of this design.
>
>
> **Training data parts:** To balance the performance across tasks, we adjust the proportion of training data for each dataset. In our setup, the training data is sampled with a ratio of 1:25:1 for COCO-panoptic, Youtube-2019, and COCO-SAM datasets, respectively. Adopting this data balance method, we can still achieve strong performance of VIS.
>
> At last, when we compare with SAM-like model, we find that co-training is not consistently effective, as performance degradation may occur due to the increased complexity and challenges of multi-task joint co-training.
> To address this, we first pretrain our model on the COCO dataset. After co-training, we finetune the model using the SAM dataset (5\%) to enhance interactive segmentation performance. Thus, the results are reported in other tables (see the Tab.4, Tab.10,11,12).
>
> **We have updated these detailed processes in our update draft, and we will open source the training code for these settings.**
>
>
>
> #### Q2： This paper ignores some related methods, making it difficult to assess the model's performance relative to existing SOTA approaches. For example, some universal methods obtain better results than RMP-SAM using ResNet50. The authors should make a comprehensive comparison with other methods.
>
> Thanks for your suggestion. We agree that the scope of our research is orthogonality to the works[1]-[3].
>
> Firstly, since our method is not designed for specific models and our method is trained on multiple purposes tasks, directly comparing our method with previous works[1][2] may not be fair. As our core goal is unifying multiple tasks in real-time scenarios and adopting a multi-task co-training strategy, our method may not perform the best on a single task.
>
> Secondly, The work[3] also designs a unified model for video segmentation tasks. However, it cannot perform SAM-like segmentation in real time. We adopt joint image-video data co-training rather than image or video data. Thus, we can keep SAM-like segmentation, panoptic segmentation, and video instance segmentation in one model and run in real time.
>
> Thirdly, all these methods focus on achieving strong results rather than real-time design. The work[2] uses extra transformer encoders after heavy Mask2Former architectures, which brings more computation costs.
>
> At last, following your suggestion, we have added a comparison with previous SOTA methods on YTVIS--2019. We follow the work[1][2], with the modification of replacing pre-training on COCO with our co-training on COCO, COCO-SAM, and YTVIS-2019.
> Then, we follow the works[1][2] and finetune the pre-trained model on YTVIS-2019.
> Nevertheless, we achieved 47.2 results when using the ResNet-50 backbone. To provide more comparison, we use ConvNeXt-L as our backbone and train a model based on the joint co-training described in the paper, achieving a result of 62.2. The results indicate our method can still achieve comparable results with these expert models.
>
>
> |Method|Backbone|YouTube-VIS 2019| COCO-SAM | COCO-Panoptic|
> |:-:|:-:|:-:|:-:|:-:|
> |UniVS| R50|47.4|-|-|
> |Dvis|R50|51.2|-|-|
> |Tube-Link|R50| 52.8|-|-|
> |RMP-SAM(Ours)|R50|47.2| 55.3 | 46.5 |
> |UniVS| Swin-L|60.0|-|-|
> |Dvis|Swin-L|63.9|-|-|
> |Tube-Link|Swin-L| 64.6|-|-|
> |RMP-SAM(Ours)|ConvNeXt-L|62.2| 60.8 | 52.0 |
>
>
> According to your suggestion, we have updated this detaild comparison in our appendix. Please check our updated draft.
>
>
> [1] Tube-Link: A flexible cross tube framework for universal video segmentation. ICCV 2023.
>
> [2] Dvis: Decoupled video instance segmentation framework. ICCV 2023.
>
> [3] Univs: Unified and universal video segmentation with prompts as queries. CVPR 2024

---

> > ### Author Response · Authors · 2024-11-25
> > **Please let us know whether all issues are addressed**
> >
> > Dear reviewer,
> >
> > Thanks for the comments. We have provided more explanations and answers to your questions. Since the deadline for discussion is Nov 26, please let us know whether we have answered all the questions. Please also consider raising the score after all issues are addressed.
> >
> > Thanks,

---

> > > ### Author Response · Authors · 2024-11-25
> > > **Please let us know whether all issues are addressed**
> > >
> > > Dear reviewer,
> > >
> > > Thanks for the comments. We have provided more explanations and answers to your questions. We have followed your suggestions to compare more recent STOA video segmentation methods. Moreover, we also provide more details on the joint co-training.
> > >
> > > If you have further question, please ask us and we will reply it as soon as possible.
> > >
> > > Thanks,

---

> > > > ### Author Response · Authors · 2024-11-25
> > > > **Whether the questions are solved**
> > > >
> > > > Dear reviewer NMA5:
> > > >
> > > > We have updated the response and corresponding draft. Moreover, two reviewers have stated that their concerns are solved.
> > > > We want to know whether your concerns are solved, since the deadline for discussion is Nov 26.
> > > >
> > > > Best regards!
> > > >
> > > > Authors of RMP-SAM

---

> > ### Comment · Reviewer_NMA5 · 2024-11-25
> > **Official Comment by Reviewer NMA5**
> >
> > Thank you for the rebuttal. I decide to raise my score. And I hope the author can add these details in the updated version, which will be helpful for understanding and reproduction.

---

### Official Review · Reviewer_mNKa · 2024-11-02

**Soundness:** 3
**Presentation:** 3
**Contribution:** 3
**Rating:** 8
**Confidence:** 4

**Summary:**

This work addresses the need for real-time multi-purpose segmentation by introducing a novel setting that encompasses interactive, panoptic, and video instance segmentation, striving for a single end-to-end model capable of handling all tasks in real-time. The proposed Real-Time Multi-Purpose SAM (RMP-SAM) utilizes an efficient encoder and a decoupled adapter for prompt-driven decoding, along with innovative training strategies and adapter designs, demonstrating effectiveness and strong generalization across benchmarks and specific semantic tasks while achieving an optimal balance between accuracy and speed.

**Strengths:**

1.Demonstrates impressive performance and inference speed.

2.Filling the gap in real-time multi-purpose segmentation.

3.The whole method is very simple and easy to understand.

4.Code is provided for easy reproduction by the reader.

**Weaknesses:**

1.Based on existing technology development, the entire pipeline is not novel.

2.Differences with SAMv2 should be further clarified, especially in terms of claimed semantic labels?

**Questions:**

See weakness.

---

> ### Author Response · Authors · 2024-11-22
> **Response to Reviewer mNKa**
>
> #### Q1: Novelty and technical contribution.
>
> Our main contribution is introducing a new setting that supports various segmentation tasks within a single real-time model. To our knowledge, there are no previous works in this direction. Most efficient models[1]-[3] are working on a single task and verifying their method on a single dataset.
>
> Thus, we benchmark several existing real-time segmentation models (including Mask2Former), extending them to handle panoptic, video instance, and interactive segmentation within one unified framework. However, most works cannot achieve the best speed and accuracy trade-off on multiple tasks.
>
> Next, we present our model, RMP-SAM, a dynamic convolution-based approach with an improved task adaptation design. Our key designs include a shared convolution-based decoder with mask pooling to accelerate the decoding process and a decoupled adapter to decode the semantic-aware masks (image masks, video tube masks) and visual prompt-aware masks (interactive masks). With these designs, our method, RMP-SAM, can achieve the speed and accuracy trade-off on our proposed benchmark.
>
> RMP-SAM achieves the best trade-off between performance, task versatility, and speed. We note that expanding the image along the time dimension is merely a specific implementation strategy and not the core contribution of our work.
>
>
> [1] ICNet for real-time semantic segmentation on high-resolution images, ECCV-2018.
>
> [2] You Only Segment Once: Towards Real-Time Panoptic Segmentation, CVPR-2023.
>
> [3] Faster Segment Anything: Towards Lightweight SAM for Mobile Applications, arxiv-2023
>
>
> #### Q2: Differences with SAMv2 should be further clarified, especially in terms of claimed semantic labels?
>
> Thanks for your suggestion. Our updated draft provides a more detailed comparison with SAMv2. Here are detailed comparisons:
>
> (1) For functionality, SAM-2 mainly focuses on video object segmentation and interactive segmentation. Compared with SAM, it adds mask tracking ability on given visual prompts, while our method mainly explores multi-purpose and multi-task real-time segmentation. We unify panoptic segmentation, video instance segmentation, and interactive segmentation in one model with the requirements in real time.
>
> (2) For data scale and diversity, SAM-2 builds a large scale dataset and mainly has one purpose: Video Object Segmentation.
> Our RMP-SAM only involves a small set of public data sources and has multiple purposes: segmentation, mask labeling, mask tracking, and panoptic segmentation.
>
> (3) For goals, SAM-2 aims at the production level with large-scale datasets (including internal datasets) co-training. Our RMP-SAM aims at efficient model design and performs well under real-time constraints.
>
> (4) Last, our work is concurrent with SAM-2 since our work is also inspired by pioneering SAM.
>
> We have updated these responses in our refined draft.

---

> > ### Author Response · Authors · 2024-11-25
> > **Please let us know whether all issues are addressed**
> >
> > Dear reviewer,
> >
> > Thanks for the comments. We have provided more explanations and answers to your questions. Since the deadline for discussion is Nov 26, please let us know whether we have answered all the questions. Please also consider raising the score after all issues are addressed.
> >
> > Thanks,

---

> > ### Comment · Reviewer_mNKa · 2024-11-25
> > **Thanks to the author for the reply**
> >
> > Thanks to the author for the reply.
> >
> > Given the expanded functionality and engineering value of RMP-SAM, I think it's worth being accepted.

---

> > > ### Author Response · Authors · 2024-11-25
> > > **Thank you**
> > >
> > > Dear reviewer,
> > >
> > > Thanks for letting us know all the questions have been answered. Please consider increasing your rating as you think this paper is worth being accepted.
> > >
> > > Thanks,

---

### Official Review · Reviewer_ycmt · 2024-11-04

**Soundness:** 3
**Presentation:** 3
**Contribution:** 3
**Rating:** 8
**Confidence:** 3

**Summary:**

The paper presents a real-time, versatile segmentation model capable of interactive segmentation, panoptic segmentation, and video instance segmentation.
While retaining the SAM encoder-decoder structure, the model incorporates an efficient encoder and adapter to enhance performance.
In the decoder, RAP-SAM introduces a three-stage pipeline that leverages novel pooling-based dynamic convolutions to refine mask tokens. Following the decoder, two additional prompt adapters are implemented to improve interaction between visual prompts and segmentation tokens.
RAP-SAM demonstrates efficiency and generalizability across various segmentation benchmarks.

**Strengths:**

1. The model achieves multi-purpose segmentation through an efficient structure and unified training approach.
2. The paper is well-written and easy to follow.
3. The experiments on panoptic segmentation, interactive segmentation, and video segmentation are solid, comprehensive, and persuasive, effectively demonstrating the model's contribution.

**Weaknesses:**

1. The paper lacks a detailed comparison with other SAM-like methods. A single COCO instance segmentation comparison in Table 4 is insufficient to substantiate claims of superiority over SAM. The results presented in Table 4 are not particularly outstanding. Additional experiments, such as on the SegAny task, with detailed metrics (AP for small, medium, large objects) on COCO instance segmentation, and evaluations with different object detectors, would strengthen the case.

2. Efficiency benchmarks are insufficiently detailed. For a model promoting efficiency, there should be a more comprehensive evaluation across different GPU platforms, such as the 3090 and V100, testing throughput and latency. Additionally, plotting latency versus performance compared to other SAM-like methods would provide a clearer visualization of the model's efficiency.

**Questions:**

1. Do you have the results for TopFormer in Table 3? Additionally, please bold the results in all comparison tables for clarity.

---

> ### Author Response · Authors · 2024-11-22
> **Response to Reviewer ycmt**
>
> #### Q1: Compared with other SAM-like methods with more detailed metrics and more stronger detectors.
>
> Thank you for your suggestion. In the updated draft, we have provided a more detailed comparison with other SAM-like methods in Table 10-12 and Figure 5. Please check our paper for a more detailed comparison.
> The results show that our model achieves comparable or better performance across various detectors and metrics while significantly reducing FLOPs. Moreover, our method is able to support panoptic segmentation and video instance segmentation, which other SAM-like methods cannot, highlighting our core contribution: a multiple-purpose real-time segmentation model.
>
>
> #### Q2: Testing the efficiency across different GPU platforms.
>
> We have evaluated several methods from Table 2 across multiple GPU platforms, including A100-40G, A10-22G, and 3090-24G. Unfortunately, we currently do not have access to a V100 GPU and are unable to provide its corresponding results.
>
> All model results for each specific GPU were generated on the same machine. The FPS and GFlops values were calculated using an image with a 1333 x 800 pixels resolution. We report these results with the ResNet-18 backbone.
>
> |Method|GPU|FLOPs |Parameters| FPS|
> |:-:|:-:|:-:|:-:|:-:|
> |Mask2Former|A100-40G|89.8G|18.6M |31.2|
> |kMaX-DeepLab |A100-40G|87.1G |18.7M |15.0|
> |YOSO|A100-40G| 57.3G |18.7M |41.0|
> |**RMP-SAM** |**A100-40G**|**60.5G** |**22.8M**| **40.3**|
> |Mask2Former|A10-22G|89.8G|18.6M |10.1|
> |kMaX-DeepLab |A10-22G|87.1G |18.7M |4.3|
> |YOSO|A10-22G| 57.3G |18.7M |13.6|
> |**RMP-SAM**|**A10-22G**|**60.5G** |**22.8M**| **14.2**|
> |Mask2Former|3090-24G|89.8G|18.6M |25.6|
> |kMaX-DeepLab |3090-24G|87.1G |18.7M |9.0|
> |YOSO|3090-24G| 57.3G |18.7M |31.4|
> |**RMP-SAM**|**3090-24G**|**60.5G** |**22.8M**| **32.0**|
>
> As shown in the above table, our method can achieve faster speeds at A10 and 3090 and comparable speeds at A100-40G.
>
>
> #### Q3: The latency visualization results compared with other SAM-like methods.
>
> Thanks for your suggestion. We have provided visualization results in our updated draft. Please check the Figure 5 in the appendix for this. In particular, we report the GFlops, parameters, and performance comparison with these methods.
>
>
> #### Q4: The result of TopFormer in Tab-3 and bold the results in all comparison tables.
>
> Thanks for your suggestion. We have provided the results of the TopFormer and modified the table's structure by bolding some of the results for easier reading and comparison. Please revisit our paper to review these changes.

---

> > ### Author Response · Authors · 2024-11-25
> > **Please let us know whether all issues are addressed**
> >
> > Dear reviewer,
> >
> > Thanks for the comments. We have provided more explanations and answers to your questions. Since the deadline for discussion is Nov 26, please let us know whether we have answered all the questions. Please also consider raising the score after all issues are addressed.
> >
> > Thanks,

---

> > > ### Author Response · Authors · 2024-11-25
> > > **Please let us know whether all issues are addressed**
> > >
> > > Dear reviewer:
> > >
> > > We have updated the response and corresponding draft. Moreover, two reviewers have stated that their concerns are solved and one has improved his score. We want to know whether your concerns are solved and whether you can improve your score, since the deadline for discussion is Nov 26.
> > >
> > > **If you have more questions, we will reply it as soon as possible**.
> > >
> > > Best regards!
> > >
> > > Authors of RMP-SAM

---

> > > ### Comment · Reviewer_ycmt · 2024-11-25
> > > **My concerns have been fully addressed.**
> > >
> > > Thank you to the authors for their response. After reviewing the feedback from the other reviewer, I believe the work is worthy of acceptance

---

> > > > ### Author Response · Authors · 2024-11-25
> > > > **Thanks**
> > > >
> > > > Dear reviewer,
> > > >
> > > > Thanks for raising the score. We have merged your comments into the latest version.
> > > >
> > > > Best Regards!
> > > >
> > > > Authors of RMP-SAM

---

### Author Response · Authors · 2024-11-22
**General Response**

### General Response

We thank all reviewers for their valuable comments and constructive suggestions.

#### General Questions


1, Detailed Comparison with recent efficient SAM-like models.

In the previous version, we only report our model results in Tab.4, with one detector as the box prompts. Following the reviewers' suggestion, we add a more detailed comparison with these efficient SAM-like models, including more detectors and detailed metrics results on COCO-instance segmentation and more devices for speed testing. Our RMP-SAM can achieve stronger results compared with those specialists.


2, Contributions and Novelty.


Our main contribution is introducing a **new setting** that supports various segmentation tasks within a single real-time model. To our knowledge, there are no previous works in this direction. Most efficient models[1]-[3] are working on a single task and verifying their method on a single dataset.

Thus, we benchmark several existing real-time segmentation models (including Mask2Former), extending them to handle panoptic, video instance, and interactive segmentation within one unified framework. However, most works cannot achieve the best speed and accuracy trade-off on multiple tasks.

Next, we present our model, RMP-SAM, a dynamic convolution-based approach with an improved task adaptation design. Our key designs include a shared convolution-based decoder with mask pooling to accelerate the decoding process and a decoupled adapter to decode the semantic-aware masks (image masks, video tube masks) and visual prompt-aware masks (interactive masks). With these designs, our method, RMP-SAM, can achieve the speed and accuracy trade-off on our proposed benchmark.
We also verify the effectiveness of decoupled adapters on various methods and balanced results for COCO panoptic and COCO-SAM segmentation.

For the results, RMP-SAM achieves the best trade-off between performance, task versatility, and speed.


3, comparison with stronger models, SAM-2, and other strong video segmentation methods.


Following the reviewers' suggestions, we have compared with recent video segmentation methods, including SAM-2, strong baselines (DVIS and Tube-Link).

Compared with these foundation models, our model is more **efficient, multi-purposes** (both image/video, interactive/instance/panoptic in one small/efficient model, see the Tab.1 and Tab.2 in our paper). The goal of our work is orthogonal to these works, and our work is fundamentally different from these works. We present a more detailed comparison of these methods.


4, Training setup.

We adopt joint image and video co-training, rather than first pre-training on image and then fine-tuning on video. We present a more detailed implementation and data-balancing strategy in the appendix. We will opensource our codebase and model for the community.


##### Summary of Changes

We've made the revisions to the main paper according to all reviewers' comments. The main revisions are summarized as follows:

1. We have revised the structure of Table 2 and bolded some information for easy reading and comparison.

2. We have added the results of RMP-SAM using TopFormer as the backbone for a more detailed comparison.

3. We have compared our method with other SAM-like methods with more detailed metrics and stronger detectors, see the appendix of Tab.10-Tab.12.

4. We have compared our method with SAM-2 in the appendix.

5. We have compared our method with recent strong video segmentation methods in the appendix.

6. We have made the reference to the appendix more precise.


The details of the revisions and other supplemented experiment results are in the following official comments to each reviewer.



[1] Tube-Link: A flexible cross tube framework for universal video segmentation. ICCV 2023.

[2] Dvis: Decoupled video instance segmentation framework. ICCV 2023.

[3] Univs: Unified and universal video segmentation with prompts as queries. CVPR 2024

---

### Comment · Area_Chair_yv4z · 2024-11-24

Dear Reviewers,

The discussion with the authors will conclude soon. The authors have provided detailed rebuttals. If there are any points that you feel have not been adequately clarified or if there are misunderstandings in their responses, please take this opportunity to raise them now. Thank you for your contributions to this review process.

---

### Meta-Review · Area_Chair_yv4z · 2024-12-21

**Metareview:**

This paper presents RAP-SAM, a real-time, versatile segmentation model capable of handling interactive segmentation, panoptic segmentation, and video instance segmentation. The model retains the SAM encoder-decoder structure while incorporating an efficient encoder and adapter to enhance performance. In the decoder, RAP-SAM introduces a three-stage pipeline leveraging pooling-based dynamic convolutions to refine mask tokens. Additionally, two prompt adapters are implemented to improve interactions between visual prompts and segmentation tokens. RAP-SAM demonstrates strong efficiency and generalizability across various segmentation benchmarks, filling the gap in real-time multi-purpose segmentation.

The reviewers unanimously praised the paper for its innovative approach to multi-purpose segmentation, including:
1.The ability to perform interactive segmentation, panoptic segmentation, and video instance segmentation through a unified training approach.
2.A well-written and easy-to-follow presentation.
3.Comprehensive experiments on multiple segmentation benchmarks, demonstrating the model's contributions and performance.
4.Strong inference speed and efficiency, meeting real-time requirements.



Therefore, I recommend the acceptance based on the unanimous support from reviewers. Since this research fills the gap in real-time multi-purpose segmentation and achieved impressive performance and inference speed, I would like to recommend for an oral presentation.

**Additional Comments On Reviewer Discussion:**

Initial concerns were raised regarding The lack of comparisons with SAM-like methods or the need for a comprehensive evaluation across different GPU platforms. However, the authors addressed these concerns thoroughly during the rebuttal stage, resolving all major points raised by the reviewers.

---

### Decision · Program_Chairs · 2025-01-22

Accept (Oral)